# Action Guidance: Getting the Best of Sparse Rewards and Shaped Rewards for Real-time Strategy Games

## Abstract

Training agents using Reinforcement Learning in games with sparse rewards is a challenging problem, since large amounts of exploration are required to retrieve even the first reward. To tackle this problem, a common approach is to use reward shaping to help exploration. However, an important drawback of reward shaping is that agents sometimes learn to optimize the shaped reward instead of the true objective. In this paper, we present a novel technique that we call *action guidance* that successfully trains agents to eventually optimize the true objective in games with sparse rewards while maintaining most of the sample efficiency that comes with reward shaping. We evaluate our approach in a simplified real-time strategy (RTS) game simulator called $\mu$RTS.

Training agents using Reinforcement Learning with sparse rewards is often difficult (Pathak et al., 2017). First, due to the sparsity of the reward, the agent often spends the majority of the training time doing inefficient exploration and sometimes not even reaching the first sparse reward during the entirety of its training. Second, even if the agents have successfully retrieved some sparse rewards, performing proper credit assignment is challenging among complex sequences of actions that have led to theses sparse rewards. Reward shaping (Ng et al., 1999) is a widely-used technique designed to mitigate this problem. It works by providing intermediate rewards that lead the agent towards the sparse rewards, which are the true objective. For example, the sparse reward for a game of Chess is naturally +1 for winning, -1 for losing, and 0 for drawing, while a possible shaped reward might be +1 for every enemy piece the agent takes. One of the critical drawbacks for reward shaping is that the agent sometimes learns to optimize for the shaped reward instead of the real objective. Using the Chess example, the agent might learn to take as many enemy pieces as possible while still losing the game. A good shaped reward achieves a nice balance between letting the agent find the sparse reward and being too shaped (so the agent learns to just maximize the shaped reward), but this balance can be difficult to find.

In this paper, we present a novel technique called *action guidance* that successfully trains the agent to eventually optimize over sparse rewards while maintaining most of the sample efficiency that comes with reward shaping. It works by constructing a *main policy* that only learns from the sparse reward function $R_{\mathcal{M}}$ and some *auxiliary policies* that learn from the shaped reward function $R_{\mathcal{A}_1}, R_{\mathcal{A}_2}, \ldots, R_{\mathcal{A}_n}$. During training, we use the same rollouts to train the main and auxiliary policies and initially set a high-probability of the main policy to take *action guidance* from the auxiliary policies, that is, *the main policy will execute actions sampled from the auxiliary policies*. Then the main policy and auxiliary policies are updated via off-policy policy gradient. As the training goes on, the main policy will get more independent and execute more actions sampled from its own policy. Auxiliary policies learn from shaped rewards and therefore make the training sample-efficient, while the main policy learns from the original sparse reward and therefore makes sure that the agents will eventually optimize over the true objective. We can see action guidance as combining reward shaping to train auxiliary policies interlieaved with a sort of imitation learning to guide the main policy from these auxiliary policies.

We examine action guidance in the context of a real-time strategy (RTS) game simulator called $\mu$RTS for three sparse rewards tasks of varying difficulty. For each task, we compare the performance of training agents with the sparse reward function $R_{\mathcal{M}}$, a shaped reward function $R_{\mathcal{A}_1}$, and action guidance with a singular auxiliary policy learning from $R_{\mathcal{A}_1}$. The main highlights are:

**Action guidance is sample-efficient.** Since the auxiliary policy learns from $R_{\mathcal{A}_1}$ and the main policy takes action guidance from the auxiliary policy during the initial stage of training, the main policy is more likely to discover the first sparse reward more quickly and learn more efficiently. Empirically, action guidance reaches almost the same level of sample efficiency as reward shaping in all of the three tasks tested.

**The true objective is being optimized.** During the course of training, the main policy has never seen the shaped rewards. This ensures that the main policy, which is the agent we are really interested in, is always optimizing against the true objective and is less biased by the shaped rewards. As an example, Figure 1 shows that the main policy trained with action guidance eventually learns to win the game as fast as possible, even though it has only learned from the match outcome reward (+1 for winning, -1 for losing, and 0 for drawing). In contrast, the agents trained with reward shaping learn more diverse sets of behaviors which result in high shaped reward.

To support further research in this field, we make our source code available at GitHub[1], as well as all the metrics, logs, and recorded videos[2].

# 1   RELATED WORK

In this section, we briefly summarize the popular techniques proposed to address the challenge of sparse rewards.

**Reward Shaping.** Reward shaping is a common technique where the human designer uses domain knowledge to define additional intermediate rewards for the agents. Ng et al. (1999) show that a slightly more restricted form of state-based reward shaping has better theoretical properties for preserving the optimal policy.

**Transfer and Curriculum Learning.** Sometimes learning the target tasks with sparse rewards is too challenging, and it is more preferable to learn some easier tasks first. *Transfer learning* leverages this idea and trains agents with some easier source tasks and then later transfer the knowledge through value function (Taylor et al., 2007) or reward shaping (Svetlik et al., 2017). *Curriculum learning* further extends transfer learning by automatically designing and choosing a full sequences of source tasks (i.e. a curriculum) (Narvekar & Stone, 2018).

**Imitation Learning.** Alternatively, it is possible to directly provide examples of human demonstration or expert replay for the agents to mimic via Behavior Cloning (BC) (Bain & Sammut, 1995), which uses supervised learning to learn a policy given the state-action pairs from expert replays. Alternatively, Inverse Reinforcement Learning (IRL) (Abbeel & Ng, 2004) recovers a reward function from expert demonstrations to be used to train agents.

**Curiosity-driven Learning.** Curiosity driven learning seeks to design *intrinsic* reward functions (Burda et al., 2019) using metrics such as prediction errors (Houthooft et al., 2016) and "visit counts" (Bellemare et al., 2016; Lopes et al., 2012). These intrinsic rewards encourage the agents to explore unseen states.

**Goal-oriented Learning.** In certain tasks, it is possible to describe a goal state and use it in conjunction with the current state as input (Schaul et al., 2015). Hindsight experience replay (HER) (Andrychowicz et al., 2017) develops better utilization of existing data in experience replay by replaying each episode with different goals. HER is shown to be an effective technique in sparse rewards tasks.

**Hierarchical Reinforcement Learning (HRL).** If the target task is difficult to learn directly, it is also possible to hierarchically structure the task using experts' knowledge and train hierarchical agents, which generally involves a main policy that learns abstract goals, time, and actions, as well as auxiliary policies that learn primitive actions and specific goals (Dietterich, 2000). HRL is especially popular in RTS games with combinatorial action spaces (Pang et al., 2019; Ye et al., 2020).

The most closely related work is perhaps Scheduled Auxiliary Control (SAC-X) (Riedmiller et al., 2018), which is an HRL algorithm that trains auxiliary policies to perform primitive actions with

---

[1]`https://github.com/anonymous-research-code/action-guidance`
[2]Blinded for peer review

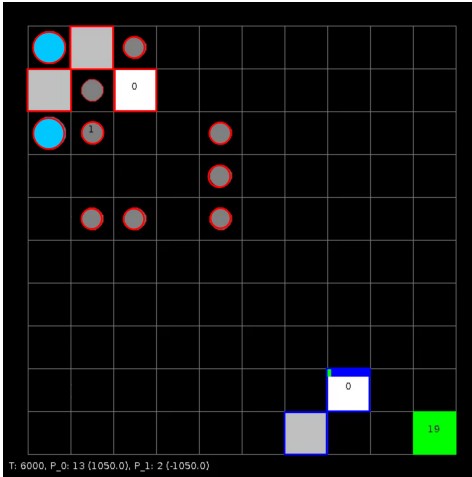

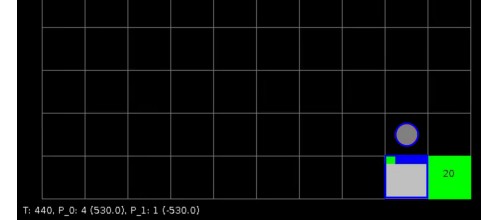

(a) shaped reward
(`https://streamable.com/o797ca`)

(b) action guidance
(`https://streamable.com/hh7abp`)

Figure 1: The screenshot shows the typical learned behavior of agents in the task of DefeatRandomEnemy. (a) shows that an agent trained with some shaped reward function $R_{\mathcal{A}_1}$ learns many helpful behaviors such as building workers (grey circles), combat units (blue circles), and barracks (grey square) or using owned units (with red boarder) to attack enemy units (with blue border), but does not learn to win as fast as possible (i.e. it still does not win at internal time step $t = 6000$). In contrast, (b) shows an agent trained with action guidance optimizes over the match outcome and learns to win as fast as possible (i.e. about to win the game at $t = 440$), with its main policy learning from the match outcome reward function $R_{\mathcal{M}}$ and a singular auxiliary policy learning from the same shaped reward function $R_{\mathcal{A}_1}$. Click on the link below figures to see the full videos of trained agents.

shaped rewards and a main policy to schedule the use of auxiliary policies with sparse rewards. However, our approach differs in the treatment of the main policy. Instead of learning to *schedule* auxiliary policies, our main policy learns to act in the entire action space by *taking action guidance* from the auxiliary policies. There are two intuitive benefits to our approach since our main policy learns in the full action space. First, during policy evaluation our main policy does not have to commit to a particular auxiliary policy to perform actions for a fixed number of time steps like it is usually done in SAC-X. Second, learning in the full action space means the main policy will less likely suffer from the definition of hand-crafted sub-tasks, which could be incomplete or biased.

## 2 BACKGROUND

We consider the Reinforcement Learning problem in a Markov Decision Process (MDP) denoted as $(S, A, P, \rho_0, r, \gamma, T)$, where $S$ is the state space, $A$ is the discrete action space, $P : S \times A \times S \to [0, 1]$ is the state transition probability, $\rho_0 : S \to [0, 1]$ is the the initial state distribution, $r : S \times A \to \mathbb{R}$ is the reward function, $\gamma$ is the discount factor, and $T$ is the maximum episode length. A stochastic policy $\pi_\theta : S \times A \to [0, 1]$, parameterized by a parameter vector $\theta$, assigns a probability value to an action given a state. The goal is to maximize the expected discounted return of the policy:

$$\mathbb{E}_\tau \left[ \sum_{t=0}^{T-1} \gamma^t r_t \right], \quad \begin{array}{l} \text{where } \tau \text{ is the trajectory } (s_0, a_0, r_0, s_1, \ldots, s_{T-1}, a_{T-1}, r_{T-1}) \\ \text{and } s_0 \sim \rho_0, s_t \sim P(\cdot|s_{t-1}, a_{t-1}), a_t \sim \pi_\theta(\cdot|s_t), r_t = r(s_t, a_t) \end{array}$$

**Policy Gradient Algorithms.** The core idea behind policy gradient algorithms is to obtain the *policy gradient* $\nabla_\theta J$ of the expected discounted return with respect to the policy parameter $\theta$. Doing gradient ascent $\theta = \theta + \nabla_\theta J$ therefore maximizes the expected discounted reward. Earlier work proposes the following policy gradient estimate to the objective $J$ (Sutton & Barto, 2018):

$$g_{\text{policy}, \theta} = \mathbb{E}_{\tau \sim \pi_\theta} \left[ \sum_{t=0}^{T-1} \nabla_\theta \log \pi_\theta(a_t|s_t) G_t \right],$$

where $G_t = \sum_{k=0}^{\infty} \gamma^k r_{t+k}$ denotes the discounted return following time $t$. This gradient estimate, however, suffers from large variance (Sutton & Barto, 2018) and the following gradient estimate is suggested instead:

$$g_{\text{policy},\theta} = \mathbb{E}_\tau \left[ \nabla_\theta \sum_{t=0}^{T-1} \log \pi_\theta(a_t|s_t) A(\tau, V, t) \right],$$

where $A(\tau, V, t)$ is the General Advantage Estimation (GAE) (Schulman et al., 2015), which measures "how good is $a_t$ compared to the usual actions", and $V : S \to \mathbb{R}$ is the state-value function.

## 3 ACTION GUIDANCE

The key idea behind *action guidance* is to create a main policy that trains on the sparse rewards, and creating some auxiliary policies that are trained on shaped rewards. During the initial stages of training, the main policy has a high probability to take *action guidance* from the auxiliary policies, that is, the main policy can execute actions sampled from the auxiliary policies, rather than from its own policy. As the training goes on, this probability decreases, and the main policy executes more actions sampled from its own policy. During training, the main and auxiliary policies are updated via off-policy policy gradient. Our use of auxiliary policies makes the training sample-efficient, and our use of the main policy, who only sees its own sparse reward, makes sure that the agent will eventually optimize over the true objective of sparse rewards. In a way, *action guidance* can be seen as training agents using shaped rewards, while having the main policy learn by imitating from them.

Specifically, let us define $\mathcal{M}$ as the MDP that the main policy learns from and $\mathcal{A} = \{\mathcal{A}_1, \mathcal{A}_2, ..., \mathcal{A}_k\}$ be a set of auxiliary MDPs that the auxiliary policies learn from. In our constructions, $\mathcal{M}$ and $\mathcal{A}$ share the same state, observation, and action space. However, the reward function for $\mathcal{M}$ is $R_\mathcal{M}$, which is the sparse reward function, and reward functions for $\mathcal{A}$ are $R_{\mathcal{A}_1}, ..., R_{\mathcal{A}_k}$, which are the shaped reward functions. For each of these MDPs $\mathcal{E} \in \mathcal{S} = \{\mathcal{M}\} \cup \mathcal{A}$ above, let us initialize a policy $\pi_{\theta_\mathcal{E}}$ parameterized by parameters $\theta_\mathcal{E}$, respectively. Furthermore, let us use $\pi_\mathcal{S} = \{\pi_{\theta_\mathcal{E}} | \mathcal{E} \in \mathcal{S}\}$ to denote the set of these initialized policies.

At each timestep $t$, let us use some exploration strategy $S$ that selects a policy $\pi_b \in \pi_\mathcal{S}$ to sample an action $a_t$ given $s_t$. At the end of the episode, each policy $\pi_\theta \in \pi_\mathcal{S}$ can be updated via its off-policy policy gradient (Degris et al., 2012; Levine et al., 2020):

$$\mathbb{E}_{\tau \sim \pi_{\theta_b}} \left[ \left( \prod_{t=0}^{T-1} \frac{\pi_\theta(a_t|s_t)}{\pi_{\theta_b}(a_t|s_t)} \right) \sum_{t=0}^{T-1} \nabla_\theta \log \pi_\theta(a_t|s_t) A(\tau, V, t) \right] \tag{1}$$

When $\pi_\theta = \pi_{\theta_b}$, the gradient in Equation 1 means on-policy policy gradient update for $\pi_\theta$. Otherwise, the objective means off-policy policy gradient update for $\pi_\theta$.

### 3.1 PRACTICAL ALGORITHM

The gradient in Equation 1 is unbiased, but its product of importance sampling ratio $\left( \prod_{t=0}^{T-1} \frac{\pi_\theta(a_t|s_t)}{\pi_{\theta_b}(a_t|s_t)} \right)$ is known to cause high variance (Wang et al., 2016). In practice, we clip the gradient the same way as Proximal Policy Gradient (PPO) (Schulman et al., 2017):

$$L^{CLIP}(\theta) = \mathbb{E}_{\tau \sim \pi_{\theta_b}} \left[ \sum_{t=0}^{T-1} \left[ \nabla_\theta \min \left( \rho_t(\theta) A(\tau, V, t), \text{clip}\left( \rho_t(\theta), \varepsilon \right) A(\tau, V, t) \right) \right] \right] \tag{2}$$

$$\rho_t(\theta) = \frac{\pi_\theta(a_t|s_t)}{\pi_{\theta_b}(a_t|s_t)}, \quad \text{clip}\left( \rho_t(\theta), \varepsilon \right) = \begin{cases} 1 - \varepsilon & \text{if } \rho_t(\theta) < 1 - \varepsilon \\ 1 + \varepsilon & \text{if } \rho_t(\theta) > 1 + \varepsilon \\ \rho_t(\theta) & \text{otherwise} \end{cases}$$

During the optimization phase, the agent also learns the value function and maximize the policy's entropy. We therefore optimize the following joint objective for each $\pi_\theta \in \pi_\mathcal{S}$:

$$L^{CLIP}(\theta) = L^{CLIP}(\theta) - c_1 L^{VF}(\theta) + c_2 S[\pi_{\theta_b}], \tag{3}$$

where $c_1, c_2$ are coefficients, $S$ is an entropy bonus, and $L^{VF}$ is the squared error loss for the value function associated with $\pi_\theta$ as done by Schulman et al. (2017). Although action guidance can be

configured to leverage multiple auxiliary policies that learn diversified reward functions, we only use one auxiliary policy for the simplicity of experiments. In addition, we use $\epsilon$-greedy as the exploration strategy $S$ for determining the behavior policy. That is, at each timestep $t$, the behavior policy is selected to be $\pi_{\theta_{\mathcal{M}}}$ with probability $1 - \epsilon$ and $\pi_{\theta_{\mathcal{D}}}$ for $\mathcal{D} \in \mathcal{A}$ with probability $\epsilon$ (note that is $\epsilon$ is different from the clipping coefficient $\varepsilon$ of PPO). Additionally, $\epsilon$ is set to be a constant $0.95$ at start for some period of time steps (e.g. 800,000), which we refer to as the *shift* period (the time it takes to start "shifting" focus away from the auxiliary policies), then it is set to linearly decay to $\epsilon_{end}$ for some period of time steps (e.g. 1,000,000), which we refer to as the *adaptation* period (the time it takes for the main policy to fully "adapt" and become more independent). Lastly, we included a pseudocode of action guidance in Algorithm 1 at the Appendix.

## 3.2 POSITIVE LEARNING OPTIMIZATION

During our initial experiments, we found the main policy sometimes did not learn useful policies. Our hypothesis is that this was because the main policy is updated with too many trajectories with zero reward. Doing a large quantities of updates of these zero-reward trajectories actually causes the policy to converge prematurely, which is manifested by having low entropy in the action probability distribution.

To mitigate this issue of having too many zero-reward trajectories, we use a preliminary code-level optimization called Positive Learning Optimization (PLO). After collecting the rollouts, PLO works by skipping the gradient update for $\pi_{\theta_{\mathcal{E}}} \in \pi_{\mathcal{S}}$ and its value function if the rollouts contains no reward according to $R_{\mathcal{E}}$. Intuitively, PLO makes sure that the main policy learns from meaningful experience that is associated with positive rewards. To confirm its effectiveness, we provide an ablation study of PLO in the experiment section.

## 4 EVALUATION

We use $\mu$RTS[3] as our testbed, which is a minimalistic RTS game maintaining the core features that make RTS games challenging from an AI point of view: simultaneous and durative actions, large branching factors and real-time decision making. To interface with $\mu$RTS, we use gym-microrts[4] (Huang & Ontañón, 2020) to conduct our experiments. The details of gym-microrts as a RL interface can be found at Appendix A.1.

## 4.1 TASKS DESCRIPTION

We examine the three following sparse reward tasks with a range of difficulties. For each task, we compare the performance of training agents with the sparse reward function $R_{\mathcal{M}}$, a shaped reward function $R_{\mathcal{A}_1}$, and action guidance with a single auxiliary policy learning from $R_{\mathcal{A}_1}$. Here are the descriptions of these environments and their reward functions.

1. LearnToAttack: In this task, the agent's objective is to learn move to the other side of the map where the enemy units live and start attacking them. Its $R_{\mathcal{M}}$ gives a +1 reward for each valid attack action the agent issues. This is of sparse reward because the action space is so large: the agent could have build a barracks or produce a unit; it is unlikely that the agents will by chance issue lots of moving actions (out of 6 action types) with correct directions (out of 4 directions) and then start attacking. Its $R_{\mathcal{A}_1}$ gives the difference between previous and current Euclidean distance between the enemy base and its closet unit owned by the agent as the shaped reward in addition to $R_{\mathcal{M}}$.

2. ProduceCombatUnits: In this task, the agent's objective is to learn to build as many combat units as possible. Its $R_{\mathcal{M}}$ gives a +1 reward for each combat unit the agent produces. This is a more challenging task because the agent needs to learn 1) harvest resources, 2) produce barracks, 3) produce combat units once enough resources are gathered, 4) move produced combat units out of the way so as to not block the production of new combat units. Its $R_{\mathcal{A}_1}$ gives +1 for constructing every building (e.g. barracks), +1 for harvesting resources, +1 for returning resources, and +7 for each combat unit it produces.

---

[3]https://github.com/santiontanon/microrts
[4]https://github.com/vwxyzjn/gym-microrts

3. DefeatRandomEnemy: In this task, the agent's objective is to defeat a biased random bot of which the attack, harvest and return actions have 5 times the probability of other actions. Additionally, the bot subjects to the same gym-microrts' limitation (See Appendix A.2) as the agents used in our experiments. Its $R_\mathcal{M}$ gives the match outcome as the reward (-1 on a loss, 0 on a draw and +1 on a win). This is the most difficult task we examined because the agent is subject to the full complexity of the game, being required to make both macro-decisions (e.g. deciding the high-level strategies to win the game) and micro-decisions (e.g. deciding which enemy units to attack. In comparison, its $R_{\mathcal{A}_1}$ gives +5 for winning, +1 for harvesting one resource, +1 for returning resources, +1 for producing one worker, +0.2 for constructing every building, +1 for each valid attack action it issues, +7 for each combat unit it produces, and $+(0.2 * d)$ where $d$ is difference between previous and current Euclidean distance between the enemy base and its closet unit owned by the agent.

## 4.2 Agent Setup

We use PPO (Schulman et al., 2017) as the base DRL algorithm to incorporate action guidance. The details of the implementation, neural network architecture, hyperparameters, proper handling of $\mu$RTS's action space and invalid action masking (Huang & Ontañón, 2020) can be found in Appendix B. We compared the following strategies:

1. **Sparse reward (first baseline).** This agent is trained with PPO on $R_\mathcal{M}$ for each task.
2. **Shaped reward (second baseline).** This agent is trained with PPO on $R_{\mathcal{A}_1}$ for each task.
3. **Action guidance - long adaptation.** The agent is trained with PPO + action guidance with $shift = 2,000,000$ time steps, $adaptation = 7,000,000$ time steps, and $\epsilon_{end} = 0.0$
4. **Action guidance - short adaptation.** The agent is trained with PPO + action guidance with $shift = 800,000$ time steps, $adaptation = 1,000,000$ time steps, and $\epsilon_{end} = 0.0$
5. **Action guidance - mixed policy.** The agent is trained with PPO + action guidance with $shift = 2,000,000$ time steps and $adaptation = 2,000,000$ time steps, and $\epsilon_{end} = 0.5$. We call this agent "mixed policy" because it will eventually have 50% chance to sample actions from the main policy and 50% chance to sample actions from the auxiliary policy. It is effectively having mixed agent making decisions jointly.

Although it is desirable to add SAC-X to the list of strategies compared, it was not designed to handle domains with large discrete action spaces. Lastly, we also toggle the PLO option for action guidance - long adaptation, action guidance - short adaptation, action guidance - mixed policy, and sparse reward training strategies for a preliminary ablation study.

## 4.3 Experimental Results

Each of the 6 strategies is evaluated in 3 tasks with 10 random seeds. We report the results in Table 1. From here on, we use the term "sparse return" to denote the episodic return according to $R_\mathcal{M}$, and "shaped return" the episodic return according to $R_{\mathcal{A}_1}$. All the learning curves can be found in Appendix C. Below are our observations.

**Action guidance is almost as sample-efficient as reward shaping.** Since the auxiliary policy learns from $R_{\mathcal{A}_1}$ and the main policy takes a lot of action guidance from the auxiliary policy during the shift period, the main policy is more likely to discover the first sparse reward more quickly and learn more efficiently. As an example, Figure 2 demonstrates such sample-efficiency in Produce-CombatUnits, where the agents trained with sparse reward struggle to obtain the very first reward. In comparison, most action guidance related agents are able to learn almost as fast as the agents trained with shaped reward.

**Action guidance eventually optimizes the sparse reward.** This is perhaps the most important contribution of our paper. Action guidance eventually optimizes the main policy over the true objective, rather than optimizing shaped rewards. Using the ProduceCombatUnits task as an example, the agent trained with shaped reward would only start producing combat units once all the resources have been harvested, probably because the +1 reward for harvesting and returning resources are easy to retrieve and therefore the agents exploit them first. Only after these resources are exhausted would the agents start searching for other sources of rewards then learn producing combat units.

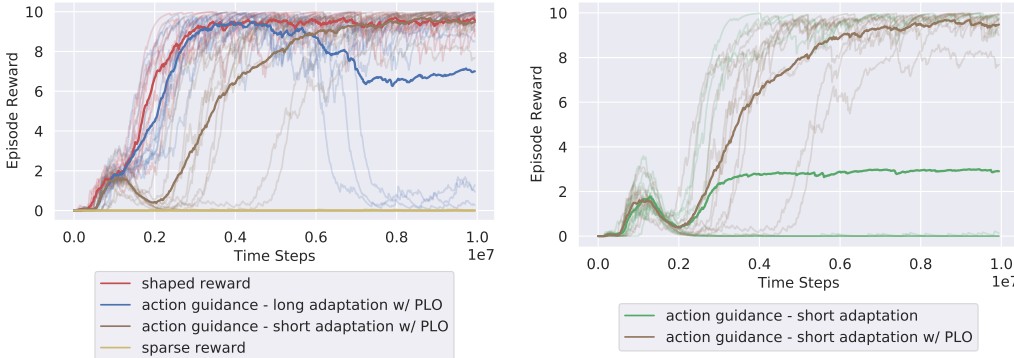

Figure 2: The faint lines are the actual sparse return of each seed for selected strategies in Produce-CombatUnits; solid lines are their means. The left figure showcase the sample-efficiency of action guidance; the right figure is a motivating example for PLO.

Table 1: The average sparse return achieved by each training strategy in each task over 10 random seeds.

|  | LearnToAttack | ProduceCombatUnit | DefeatRandomEnemy |
|---|---|---|---|
| sparse reward (first baseline) | $3.30 \pm 5.04$ | $0.00 \pm 0.01$ | $-0.07 \pm 0.03$ |
| sparse reward w/ PLO | $0.00 \pm 0.00$ | $0.00 \pm 0.01$ | $-0.05 \pm 0.03$ |
| shaped reward (second baseline) | $10.00 \pm 0.00$ | $\mathbf{9.57 \pm 0.30}$ | $0.08 \pm 0.17$ |
| action guidance - long adaptation | $\mathbf{11.00 \pm 0.00}$ | $8.31 \pm 2.62$ | $0.11 \pm 0.35$ |
| action guidance - long adaptation w/ PLO | $\mathbf{11.00 \pm 0.01}$ | $6.96 \pm 4.04$ | $\mathbf{0.52 \pm 0.35}$ |
| action guidance - mixed policy | $\mathbf{11.00 \pm 0.00}$ | $\mathbf{9.67 \pm 0.17}$ | $\mathbf{0.40 \pm 0.37}$ |
| action guidance - mixed policy w/ PLO | $10.67 \pm 0.12$ | $9.36 \pm 0.35$ | $0.30 \pm 0.42$ |
| action guidance - short adaptation | $\mathbf{11.00 \pm 0.01}$ | $2.95 \pm 4.48$ | $-0.06 \pm 0.04$ |
| action guidance - short adaptation w/ PLO | $\mathbf{11.00 \pm 0.00}$ | $\mathbf{9.48 \pm 0.51}$ | $-0.05 \pm 0.03$ |

In contrast, the main policy of action guidance - short adaptation w/ PLO are initially guided by the shaped reward agent during the shift period. During the adaptation period, we find the main policy starts to optimize against the real objective by producing the first combat unit as soon as possible. This disrupts the behavior learned from the auxiliary policy and thus cause a visible degrade in the main policy's performance during 1M and 2M timesteps as shown in Figure 2. As the adaption period comes to an end, the main policy becomes fully independent and learn to produce combat units and harvest resources concurrently. This behavior matches the common pattern observed in professional RTS game players and is obviously more desirable because should the enemy attack early, the agent will have enough combat units to defend.

In the DefeatRandomEnemy task, the agents trained with shaped rewards learn a variety of behaviors; some of them learn to do a worker rush while others learn to focus heavily on harvesting resources and producing units. This is likely because the agents could get similar level of shaped rewards despite having diverse set of behaviors. In comparison, the main policy of action guidance - long adaptation w/ PLO would start optimizing the sparse reward after the shift period ends; it almost always learns to do a worker rush, which an efficient way to win against a random enemy as shown in Figure 1.

**The hyper-parameters adaptation and shift matter.** Although the agents trained with action guidance - short adaptation w/ PLO learns the more desirable behavior, they perform considerably worse in the harder task of DefeatRandomEnemy. It suggests the harder that task is perhaps the longer adaptation should be set. However, in ProduceCombatUnits, agents trained with action guidance - long adaptation w/ PLO exhibits the same category of behavior as agents trained with shaped reward, where the agent would only start producing combat units once all the resources have been harvested. A reasonable explanation is that higher adaptation gives more guidance to the main policy to consistently find the sparse reward, but it also inflicts more bias on how the task should be accomplished; lower adaption gives less guidance but increase the likelihood for the main policy to find better ways to optimize the sparse rewards.

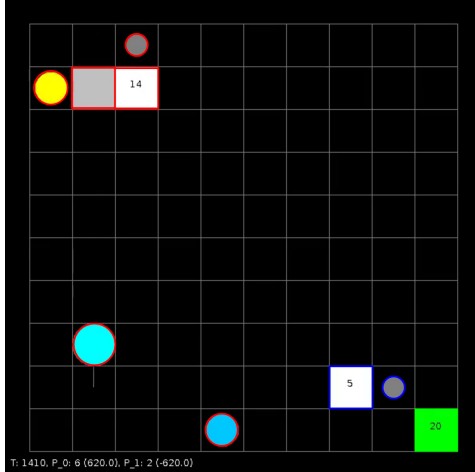 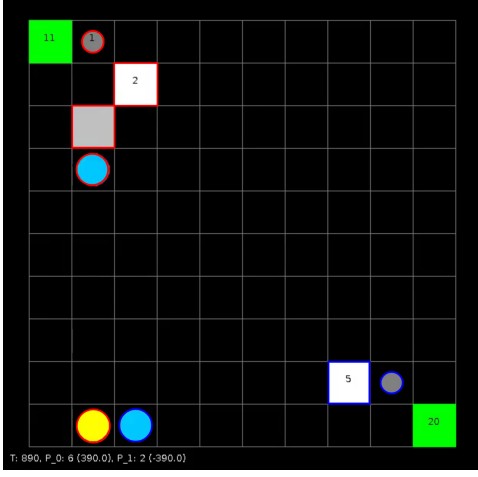

(a) shaped reward
(https://streamable.com/ytpt7u)

(b) action guidance
(https://streamable.com/mpzxef)

Figure 3: The screenshot shows the typical learned behavior of agents in the task of ProduceCombatUnits. (a) shows an agent trained with shaped reward function $R_{\mathcal{A}_1}$ learn to only produce combat units once the resources are exhausted (i.e. it produces three combat units at $t = 1410$). In contrary, (b) shows an agent trained with action guidance learn to produce units and harvest resources concurrently (i.e. it produces three combat units at $t = 890$). Click on the link below figures to see the full videos of trained agents.

**Positive Learning Optimization results show large variance.** We found PLO to be an interesting yet sometimes effective optimization in stabilizing the performance for agents trained with action guidance. However, the results show large variance: PLO either significantly helps the agents or make them much worse. As a motivating example, Figure 2 showcases the actual sparse return of 10 seeds in ProduceCombatUnits, where agents trained with action guidance - short adaptation and PLO seem to always converge while agents trained without PLO would only sometimes converge. However, PLO actually hurt the performance of action guidance - long adaptation in ProduceCombatUnits by having a few degenerate runs as shown in Figure 2. It is also worth noting the PLO does not help the sparse reward agent at all, suggesting PLO is a an optimization somewhat unique to action guidance.

**Action guidance - mixed policy is viable.** According to Table 1, agents trained with action guidance - mixed policy with or without PLO seem to perform relatively well in all three tasks examined. This is a interesting discovery because it suggests action guidance could go both ways: the auxiliary policies could also benefit from the learned policies of the main policy. An alternative perspective is to consider the main policy and the auxiliary policies as a whole entity that mixes different reward functions, somehow making joint decision and collaborating to accomplish common goals.

## 5 CONCLUSIONS

In this paper, we present a novel technique called *action guidance* that successfully trains the agent to eventually optimize over sparse rewards yet does not lose the sample efficiency that comes with reward shaping, effectively getting the best of both worlds. Our experiments with DefeatRandomEnemy in particular show it is possible to train a main policy on the full game of $\mu$RTS using only the match outcome reward, which suggests action guidance could serve as a promising alternative to the training paradigm of AlphaStar (Vinyals et al., 2019) that uses supervised learning with human replay data to bootstrap an agent. As part of our future work, we would like to scale up the approach to defeat stronger opponents.

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
