# OpenReview forum: "Action Guidance: Getting the Best of Sparse Rewards and Shaped Rewards for Real-time Strategy Games"
_ICLR.cc/2021/Conference — Reject_

### Official Review · AnonReviewer4 · 2020-10-15
**Solid paper, with some points to improve.**

**Rating:** 6
**Confidence:** 4

**Review:**

-----------------------------------------
Post-Rebuttal
-----------------------------------------
I believe both the strong points and the weaknesses I pointed out in my review remain valid. Therefore I am keeping my score unchanged.
In case of acceptance, I suggest to the authors to carefully address all the comments from my review and the other ones in the camera-ready version (especially the issues related to clarity.).

----------------------------------------
The authors propose a reward shaping method to learn faster a reinforcement learning task. Their method consists of learning multiple policies, one optimized on the real reward, while each of the other policies is optimized on a different reward shaping function.

---------------------
# Pros

- relevant research topic
- paper is well-written and generally easy to understand
- novel approach - I can't remember another approach that learns multiple policies from multiple shaping rewards and balance between them
- Experimental results seem good
- Codification freely available

--------------------------------------
# Cons

- Could have included more related works in the experimental evaluation


----------------------------
# Further comments

- I wouldn't say that you train multiple "agents", one from each reward shaping function. A more appropriate name would be saying that you train multiple "policies", each of them maximizing each of the "reward signals". Then, the connection with Probabilistic Policy Reuse would become more obvious (btw, you should discuss the relation of your work with it)

Fernández, Fernando, and Manuela Veloso. "Probabilistic policy reuse in a reinforcement learning agent." Proceedings of the fifth international joint conference on Autonomous agents and multiagent systems. 2006.

Usually, when you are trying another "agent" you are implying that this agent will explore by itself, which is not true for your method. You should read more about that (and discuss the relation with it) in the survey below:

Silva, Felipe Leno, and Anna Helena Reali Costa. "A survey on transfer learning for multiagent reinforcement learning systems." Journal of Artificial Intelligence Research 64 (2019): 645-703.

- Section 1 could be improved by more explicitly discussing the difference between the current paper and each category of related works. In special, I missed a more comprehensive discussion of reward shaping approaches and the inclusion of state-of-the-art reward shaping algorithms in the experimental evaluation.

- Why did you include a graph in Figure 2 in which "shaped reward" performed better than all others? You should perhaps also included the number of time steps used for each approach to converge to the best performance because sample complexity is as important as asymptotic performance.

- It is not very clear to me exactly how the agent selects which policy to follow at every step. Maybe it would be better to have an algorithm in the manuscript.

---

> ### Author Response · Authors · 2020-11-14
> **Thanks for the review!**
>
> We thank the reviewer for their time and constructive feedback. We especially appreciate the reviewer's kind words about the novelty of our approach, great suggestion on the term usage, and helpful references.
>
> Below we would like to address the concerns mentioned in the review:
>
> Q: I wouldn't say that you train multiple "agents", one from each reward shaping function. A more appropriate name would be saying that you train multiple "policies", each of them maximizing each of the "reward signals". Then, the connection with Probabilistic Policy Reuse would become more obvious (btw, you should discuss the relation of your work with it).
>
> A: We are very thankful for the reviewer's suggestion and helpful references. We completely agree that we should be using terms like the "main policy" and "auxiliary policy" and have updated terms in the manuscript.
>
> Q: Section 1 could be improved by more explicitly discussing the difference between the current paper and each category of related works. In special, I missed a more comprehensive discussion of reward shaping approaches and the inclusion of state-of-the-art reward shaping algorithms in the experimental evaluation.
>
> A: Given the similar comments by other reviewers, we plan to rewrite the related work section in the camera ready version, focusing more on the connection between previous sparse rewards methods and our work.
>
> Q: Could have included more related works in the experimental evaluation.
>
> A: We thank the reviewer for bringing up the need for comparing our results with other related work. Notice, however, that comparison is not as trivial as it seems, as we are interested in RTS games, and some existing algorithms would have to be adapted to work with the combinatorial action space. Moreover, our approach is orthogonal to the choice of learning algorithm, as it can be applied on top of any RL algorithm, thus we chose PPO, as just a representative algorithm.
>
> That being said, Imitation learning has been a proven approach for RTS games (Vinyals et al., 2019) that we might be able to use as a SOTA baseline for comparison. However, running experiments with StarCraft II is prohibitively expensive and there are no high-quality replay data for smaller research-oriented RTS platforms (like $\mu$RTS) to the best of our knowledge, beyond just running existing agents to generate replays.
>
> Many work in curiosity-driven Learning might make good baseline algorithms for comparison, but are often designed to deal with pixel states inputs, so it is not clear how they might work with the low-level vector states that are used for our experiments. That being said, adapting these curiosity driven learning to our set up is indeed part of our future work. We are also interested in exploring the integration of curiosity-driven learning with action guidance, for example, we could train an auxiliary agent that learns from the curiosity reward.
>
> Q: Why did you include a graph in Figure 2 in which "shaped reward" performed better than all others?
>
> A:  We use Figure 2 primarily to illustrate sample complexity: the action guidance agents' learning curves are similar to that of the shaped reward agents. Additionally, we use it to show that while the agents performance look similar in charts, they could learn drastically different behavior as per Figure 3, thus it is important to inspect the agents' actual behaviors.
>
> Q: You should perhaps also included the number of time steps used for each approach to converge to the best performance because sample complexity is as important as asymptotic performance.
>
> A: That is a great suggestion. For the camera-ready, we will add another table that shows the number of time steps it takes to converge to top returns.
>
> Q: It is not very clear to me exactly how the agent selects which policy to follow at every step. Maybe it would be better to have an algorithm in the manuscript.
>
> A: We thank the reviewer for bring up this great point. We have added the pseudocode in the Appendix for clarification.
>
> We would be happy to provide further clarification on any of these questions.

---

### Official Review · AnonReviewer3 · 2020-10-28
**This work proposed an algorithm called action guidance to solve the sparse reward problem. However, I don't think this paper is fully prepared for submission as the method is not novel enough and there exist some possible issues that need to be discussed and resolved.**

**Rating:** 4
**Confidence:** 4

**Review:**

This work proposed an algorithm called action guidance that trains the agent to eventually optimize over sparse rewards while maintaining most of the sample efficiency that comes with reward shaping. The authors examine three sparse reward tasks with a range of difficulties to prove the effectiveness of action guidance.  However, I don't think this paper is fully prepared for submission as the method is not novel enough and there exist some possible issues that need to be discussed and resolved.

Below are the detail comments.

About the method. The key idea behind action guidance is to create a main agent that trains on the sparse rewards, and creating some auxiliary agents that are trained on shaped rewards. And the main agent follows the instructions of auxiliary agents in the initial stage and the probability of it decreases during the following training. A concern is that if there exist several auxiliary agents, how do you arrange the shaped rewards to each auxiliary agent? If there is a conflict between the shaped rewards for the training and guidance of the agent, will the main agent still be trained well? Besides，the method itself is like using imitation learning to obtain initial policy parameters and continues to optimize using sparse reward, the novelty of the method is not sufficient enough.

About the experiments. The baselines use PPO to train agents with sparse rewards or shaped rewards respectively and there are no other SOTA methods designed for sparse rewards compared in the experiments, which is not convinced. Besides, in the environment ProduceCombatUnits, the shaped rewards include the reward for each combat unit the agent produces, which is exactly the sparse reward. Is it means that the agent using shaped rewards has the same optimization direction as the one using sparse rewards? I'm not sure if this is fair enough as the effectiveness of action guidance is not clear in this setting. Lastly, the random opponents in the experiments are not strong, I'm wondering about the agent's performance in a harder setting.

About the writing. The paper is well-written and self-contained. However, the figures to show the typical learned behavior of agents are not clear enough. For example, it's a little bit hard to recognize the enemy units as the blue borders are too thin.

Overall, I vote for a  rejection.

---

> ### Author Response · Authors · 2020-11-14
> **Thanks for the review**
>
> We thank the reviewer for their time and constructive feedback. We especially appreciate the review's insightful comments on balancing several auxiliary agents.
>
> Below we would like to address the concerns mentioned in the review:
>
> Q: A concern is that if there exist several auxiliary agents, how do you arrange the shaped rewards to each auxiliary agent? If there is a conflict between the shaped rewards for the training and guidance of the agent, will the main agent still be trained well?
>
> A: We appreciate the reviewer for bringing up this insightful concern. We do not currently have answers for these questions. For the simplicity of our experiments, we have simply used one auxiliary agent. In general, we suspect balancing between multiple auxiliary agents to be a non-trivial challenge. However, it is an interesting topic and we would like to study more for future work, and we will point this out in our future work section, acknowledging the lack of an answer to this question.
>
> Q: Besides, the method itself is like using imitation learning to obtain initial policy parameters and continues to optimize using sparse reward, the novelty of the method is not sufficient enough.
>
> A: The approach the reviewer mentioned (``using imitation learning to obtain initial policy parameters and continues to optimize using sparse reward'') has one key limitation: it requires high-quality initial data for the imitation learning. However, in many domains, it is not always possible to acquire this data. The novelty of our approach is that we are trading off the need of data for imitation learning with just requiring shaped reward functions to work.
>
> Q: About the experiments. The baselines use PPO to train agents with sparse rewards or shaped rewards respectively and there are no other SOTA methods designed for sparse rewards compared in the experiments, which is not convinced.
>
> A: We thank the reviewer for bringing up the need for comparing our results with other SOTA sparse reward algorithms. Notice, however, that comparison is not as trivial as it seems, as we are interested in RTS games, and some existing algorithms would have to be adapted to work with the combinatorial action space. Moreover, our approach is orthogonal to the choice of learning algorithm, as it can be applied on top of any RL algorithm, thus we chose PPO, as just a representative algorithm.
>
> That being said, Imitation learning has been a proven approach for RTS games (Vinyals et al., 2019) that we might be able to use as a SOTA baseline for comparison. However, running experiments with StarCraft II is prohibitively expensive and there are no high-quality replay data for smaller research-oriented RTS platforms (like $\mu$RTS) to the best of our knowledge, beyond just running existing agents to generate replays.
>
> Many work in curiosity-driven Learning might make good baseline algorithms for comparison, but are often designed to deal with pixel states inputs, so it is not clear how they might work with the low-level vector states that are used for our experiments. That being said, adapting these curiosity driven learning to our set up is indeed part of our future work. We are also interested in exploring the integration of curiosity-driven learning with action guidance, for example, we could train an auxiliary agent that learns from the curiosity reward.
>
> Q: Besides, in the environment ProduceCombatUnits, the shaped rewards include the reward for each combat unit the agent produces, which is exactly the sparse reward. Is it means that the agent using shaped rewards has the same optimization direction as the one using sparse rewards? I'm not sure if this is fair enough as the effectiveness of action guidance is not clear in this setting.
>
> A: As the reviewer pointed out, the shaped reward function in ProduceCombatUnits is a "superset" of the sparse reward function in this task. This exactly explains why the shaped reward agents are able to perform so well in this task, and why we think it is an interesting result that the main agent is able to perform just as well (if not better) even though it has only seen the sparse reward function.
>
> Q: Lastly, the random opponents in the experiments are not strong, I'm wondering about the agent's performance in a harder setting.
>
> A: Thanks for this comment. As part of our current work, we are starting to scale the approach up to play against harder opponents. However, notice that defeating a random opponent is not as easy a task as it seems, as for achieving that, the agent needs to learn to perform all the different game tasks, from scratch.
>
> Q: However, the figures to show the typical learned behavior of agents are not clear enough.
>
> A: We apologize for this inconvenience. All of the videos in our experiments are already recorded, but we have manually thickened the boarders of the units in the paper so they are more legible.
>
> We would be happy to provide further clarification on any of these questions.

---

### Official Review · AnonReviewer1 · 2020-10-29
**Interesting approach but somewhat immature**

**Rating:** 6
**Confidence:** 3

**Review:**

This paper introduces an approach called action guidance, made to address issues in more standard applications of reward shaping. The main idea of their approach is that there are two different kinds of agents, one (auxiliary agents) that learn from shaped reward functions alone and the other (main agent(s)) that learn only from the actual sparse rewards. The authors made use of a simplified RTS domain and demonstrated that their approach outperformed a more naive shaped reward approach. In addition they demonstrated an ablation study on positive learning optimization.

The basic idea of this paper is simple and elegant, and the paper is well-written. The results are somewhat messy, but overall represent a positive signal for this approach. The paper is also well-written and the video examples are effective at conveying the results.

I have a number of concerns with this current paper draft. First, it’s unclear to me why the authors chose this RTS environment or why the authors didn’t try several different environments to show the generality of this approach. Further, the evaluation results seem somewhat inconsistent. The action guidance agents overall do quite well but the shaped reward baseline has comparable performance on the first two tasks. This further leads me to believe that this may not be the best environment to test this approach. It’s also somewhat disappointing that the PLO results are inconclusive, this indicates that the work is perhaps still somewhat immature.

Overall I lean slightly towards acceptance. I think that the basic idea here is potentially very impactful. However, my concerns listed above hold me back from stronger support for the paper in its current state.

Questions for the authors:
1. Why did the authors choose this RTS environment?
2. How did the authors account for the performance of the shaped reward baseline for the first two tasks?
3. Do the authors not have a clearer sense of why the PLO results were inconclusive? Or what experiments could be run to delve into this further?

---

> ### Author Response · Authors · 2020-11-14
> **Thanks for the review!**
>
> We thank the reviewer for their time and constructive feedback. We especially appreciate the reviewer's kind words that our approach is simple, elegant, and potentially impactful.
>
> Below we would like to address the concerns mentioned in the review:
>
> Q: Why did the authors choose this RTS environment?
>
> A: We chose $\mu$RTS mainly due to ease of experimentation, as it is an open source engine, which we could easily modify and also set up scenarios if arbitrary complexity, from tiny to very large scenarios. Additionally, a very large set of existing agents (from annual AI competitions) are available for this environment, which make for great baselines.
>
> Q: How did the authors account for the performance of the shaped reward baseline for the first two tasks?
>
> A: Reward shaping is indeed a powerful technique when the shaped reward aligns well with the optimal policy. While this alignment happens some times, it is hard to guarantee. Thus, the downside is that the algorithm RL agents will focus on learning these good behaviors instead of learning the true objective (e.g., finishing the game as soon as possible, exactly what happened in Figure 1 (a)). We see a reflection of exactly this in our experiments, where in two of our tasks, reward shaping works very well, but not in the third. Action guidance, however, was robust in all three tasks *even though its auxiliary agent and the shaped reward agents use the same shaped reward function*.
>
> Q: Do the authors not have a clearer sense of why the PLO results were inconclusive? Or what experiments could be run to delve into this further?
>
> A: Thanks for the interest in the PLO experiments. Perhaps we should phrase the results as "PLO results show large variance" instead of "PLO results are inconclusive". As per Table 1, PLO overall either really helps the action guidance agents or makes it worse. Nevertheless, we still think PLO is a rather interesting idea following the heuristic of "skipping learning bad trajectories" or "focus on good experience". We would like to understand the situations where it works well in our future work. One potential experiment we are considering is to "soften" PLO by, rather than removing bad trajectories completely, removing them with some probability because maybe not all non-good experience is bad.
>
> We would be happy to provide further clarification on any of these questions.

---

### Official Review · AnonReviewer2 · 2020-10-29
**Interesting idea. Needs to discuss the many assumptions made by the approach.**

**Rating:** 4
**Confidence:** 4

**Review:**

The paper introduces an approach for learning policies across multiple MDPs and using those policies to improve learning performance on the task that the agent designer cares about. The approach assumes that a set of MDPs are provided to the learning agent, and that all of the MDPs have the same underlying task but with different reward densities (i.e., some of these MDPs have shaped rewards, and thus are faster to learn from). The approach operates by training the main agent to imitate the actions chosen by the other agents that are trained on the MDPs with shaped reward functions.

Overall, the approach is interesting and can be applicable to multi-task learning benchmarks, even though in its current presentation the authors do not focus on those settings.

Pros:
The paper is well-written.
The presentation of the idea is clear.
The experiment section and the results are easy to understand.

Cons:
It seems like the overall contribution seems small, as the approach assumes access to many MDPs with different reward densities. Given such an assumption, it is natural to understand why the current approach works.
The scope of the current work is limited to RTS domains.
A concise description of the algorithm seems to be missing, making it difficult to understand the overall algorithm.

Questions:
1.The approach relies on having access to many MDPs. It would be useful to describe the MDPs that the authors have considered for their experiments?
2. It seems like the different policies can be different only if their corresponding reward functions are different. Is that right? If so, in many domains, the approach relies on careful design of different MDPs to get the approach to work. The authors need to discuss these assumptions made by their approach.
3. From the experiments, it seems like the baseline agent with shaped rewards produces the same asymptotic performance as that of the agent with action guidance. If so, then the authors need to justify the use of additional computation for learning the many policies for action guidance.

---

> ### Author Response · Authors · 2020-11-14
> **Thanks for the review!**
>
> We thank the reviewer for their time and constructive feedback. We especially appreciate the reviewer's critique on our MDP formulation and careful examination of our results section.
>
> Below we would like to address the concerns mentioned in the review:
>
> Q:  It seems like the overall contribution seems small, as the approach assumes access to many MDPs with different reward densities. Given such an assumption, it is natural to understand why the current approach works... It seems like the different policies can be different only if their corresponding reward functions are different. Is that right? If so, in many domains, the approach relies on careful design of different MDPs to get the approach to work. The authors need to discuss these assumptions made by their approach. The approach relies on having access to many MDPs. It would be useful to describe the MDPs that the authors have considered for their experiments.
>
> A: We are glad that the reviewer find our approach natural to understand. However, notice that our assumptions are not that strong. We only assume that it is possible to design different reward functions for the same problem. So we are not really designing completely new MDPs, as they only vary in the reward function. Moreover, when comparing our work with existing work like AlphaStar (Vinyals et al., 2019), this requires human replay data to bootstrap the initial policy. However, it is not always possible to obtain high quality human replay data in many domains. So the novelty of our approach is that we are trading off the need of replay data (which might be hard to acquire in many domains), by just requiring designing additional reward functions for providing "action guidance".
>
> Q: A concise description of the algorithm seems to be missing, making it difficult to understand the overall algorithm.
>
> A: We thank the reviewer for bring up this great point. We have added pseudo-code in the Appendix for clarification.
>
> Q: From the experiments, it seems like the baseline agent with shaped rewards produces the same asymptotic performance as that of the agent with action guidance. If so, then the authors need to justify the use of additional computation for learning the many policies for action guidance.
>
> A: Notice that what the reviewer points out is only true for the first two tasks. However, in the third task (full game setting), shaped reward functions become more difficult to tune. If you give too many shaped rewards for good behaviors, the RL agents will focus on learning these good behaviors instead of learning the true objective of finishing the game as soon as possible (exactly what happened in Figure 1 (a)). This is exactly the weakness of reward shaping that we are trying to address: if the shaped reward aligns with the target reward, reward shaping is very powerful! However, this alignment is hard to guarantee. This is exactly echoed in our experiments. Action guidance, however, was robust in all three tasks *even though its auxiliary agent and the shaped reward agents use the same shaped reward function*.
>
> Additionally, from a computational standpoints, there is almost no practical runtime difference between the baseline agents and action guidance agents: they all take roughly 5 hours to finish training. The reason is that the RL methods are bound by 1) rollouts time, 2) inference time, 3) training time. Since action guidance agents reuse the rollouts to train both the main agent and auxiliary agent, the cost incurred at training time is rather small.
>
> We would be happy to provide further clarification on any of these questions.

---

### Decision · Program_Chairs · 2021-01-07
**Final Decision**

**Decision:**

Reject

**Comment:**

Summary:
This paper proposes an interesting idea where additional auxiliary tasks allow an agent to more quickly learn in a sparse reward task.

Comments:
* The authors may want to look at "Parallel Multi-Environment Shaping Algorithm for Complex Multi-step Task"
https://www.sciencedirect.com/science/article/abs/pii/S092523122030655X
as it has a somewhat related idea and is also in RTS games.
* It wasn't clear to me how the auxiliary tasks were generated/selected, or how the algo would work if a poor auxiliary task was used.
* I wasn't sure why SAC-X wasn't empirically compared to in a domain where it and this method could both apply.

Discussion:
The reviewers agreed that this paper could be significantly improved in multiple dimensions.

Recommendation:
I recommend we reject this paper. However, I encourage the authors to work to improve it as I'd really like to understand where and why this method is successful --- I would eventually like to incorporate it into my own work.